# Temperature-Related Short-Term Succession Events of Bacterial Phylotypes in Potter Cove, Antarctica

**DOI:** 10.3390/genes14051051

**Published:** 2023-05-08

**Authors:** Doris Ilicic, Danny Ionescu, Jason Woodhouse, Hans-Peter Grossart

**Affiliations:** 1Department of Experimental Limnology, Leibniz Institute of Freshwater Ecology and Inland Fisheries, 16775 Neuglobsow, Germany; 2Institut für Zoologie, Universität Hamburg, 20146 Hamburg, Germany; 3Institute of Biochemistry and Biology, University of Potsdam, 14469 Potsdam, Germany

**Keywords:** bacterioplankton, temperature, climate change, intraspecific variation, biogeography, bacterial community composition

## Abstract

In recent years, our understanding of the roles of bacterial communities in the Antarctic Ocean has substantially improved. It became evident that Antarctic marine bacteria are metabolically versatile, and even closely related strains may differ in their functionality and, therefore, affect the ecosystem differently. Nevertheless, most studies have been focused on entire bacterial communities, with little attention given to individual taxonomic groups. Antarctic waters are strongly influenced by climate change; thus, it is crucial to understand how changes in environmental conditions, such as changes in water temperature and salinity fluctuations, affect bacterial species in this important area. In this study, we show that an increase in water temperature of 1 °C was enough to alter bacterial communities on a short-term temporal scale. We further show the high intraspecific diversity of Antarctic bacteria and, subsequently, rapid intra-species succession events most likely driven by various temperature-adapted phylotypes. Our results reveal pronounced changes in microbial communities in the Antarctic Ocean driven by a single strong temperature anomaly. This suggests that long-term warming may have profound effects on bacterial community composition and presumably functionality in light of continuous and future climate change.

## 1. Introduction

Antarctic waters are generally poorly understood, and although the number of studies focused on marine microbial communities has increased in recent years, our knowledge of their dynamics and functional capacity is still limited [1,2].

Bacterioplankton represents the most significant amount of biomass in the Southern Ocean [3,4] and is compositionally diverse, with three major classes of bacteria present, namely *Alphaproteobacteria*, *Gammaproteobacteria,* and *Bacteroidia* (formerly Bacteroidetes) [2,5]. As in other marine ecosystems, these communities play important roles in many processes within Antarctic waters, such as primary production, degradation and turnover of organic matter, and biogeochemical cycling, which are essential components of the pelagic marine food webs [6,7,8]. Thus, it is fundamental to understand the drivers of variation in Antarctic bacterial community structure and functionality.

Temporal fluctuations in bacterioplankton community composition have been reported frequently in Antarctic waters [3,9,10,11,12] and were shown to be driven by the strong seasonal changes in sea-ice cover and light availability that characterize the Southern Ocean [12,13]. Several studies have suggested that changes in bacterial communities are strongly correlated with seasonal succession in phytoplankton communities [10,14,15,16]. In Antarctic marine environments, short-term phytoplankton blooms occur during austral spring and summer, following sea ice melt. These blooms supply marine food webs with organic carbon and nutrients, likely creating new ecological niches for heterotrophic bacteria, structuring bacterial communities, and driving the short-term shifts in bacterial community composition [10,12,17].

Most Antarctic bacteria are psychrotrophic, which means that they are well adapted, but not restricted, to cold temperatures [9,18]. Wiebe et al. [19] proposed that key factors in determining microbial distribution in Antarctic ecosystems are temperature-substrate interactions. Together, these concepts suggest that the ongoing increase in temperature potentially stimulates bacterial growth rates. However, contradictory results have also been reported, and the true effect still remains unknown [20].

Culture-independent phylogenetic studies have shown that the diverse community of Antarctic bacterioplankton shares many taxa with other oceanic systems. Nevertheless, the rapid increase in metagenomic data allows for resolving differences between phylotypes or ecotypes [21,22]. High levels of intraspecific genetic diversity and co-existence of different bacterial genotypes belonging to the same phylogenetic group have been well documented across different ocean systems [23,24,25,26], but data on microbial communities for Antarctic waters are still scarce. Landone Vescovo et al. [27] reported some bacterial phylotypes in Potter Cove (Antarctic Peninsula) that were widely divergent among themselves but also between the most closely related sequences. The latter study suggests the presence of endemic and genetically divergent genotypes, e.g., within the *Rhodobacteracea* family and the *Gammaproteobacteria* class, challenging the assumption that marine planktonic microorganisms are ubiquitously distributed [28].

Recent studies on the temporal dynamics of Antarctic bacterioplankton have mostly focused on entire bacterial assemblages, but not much work has been performed on specific taxa and their variations. Considering that Antarctic waters are highly influenced by climate change and exhibit rapid variations in environmental conditions, its microorganisms had to develop efficient adaptation strategies in order to survive. Mechanisms such as local adaptation and phenotypic plasticity have been shown to generate intraspecific variation [29]. Various traits can change rapidly within generations and differ drastically across populations in dissimilar habitats. The importance of intraspecific variation is based on the fact that its effects on community structure and trophic interactions rival those of among-species variations [30,31]. Two strains with identical 16S rRNA gene sequences can harbor divergent physiological and, consequently, ecological characteristics [32], thereby having the potential to regulate the temporal variability in biogeochemical processes differently. Identifying drivers and patterns of this genetic differentiation using multi-omics approaches is, therefore, crucial in our understanding of the Southern Ocean ecology and ecosystem function, especially in the context of current and future climate change [33,34].

In this study, we focused on the short-term temporal dynamics of bacterial communities in Potter Cove (King George Island, Antarctic Peninsula) in response to an unusual increase in temperature and seasonal input of glacier meltwater. This particular marine ecosystem is known to be strongly influenced by freshwater inputs from glacier meltwater run-off and is experiencing rapid physical climate change [35]. Thus, it is of great importance to assess the structure and potential of microbial communities as key predictors of adaptation to a changing environment.

## 2. Materials and Methods

### 2.1. Study Area and Sampling

The study was carried out at the German-Argentine Dallmann Laboratory of the Argentine Scientific Station “Dr. Carlini”, which is located at Potter Cove, King George Island/25 de Mayo I, Antarctic Peninsula (62°14′ S, 58°31′ W). Potter Cove (PC) is a semi-enclosed body of water, oriented SW-NE, with an area of approximately 7 km^2^ [36]. It is divided into an outer and an inner basin, separated by transversal sill [22]. Northern and eastern coasts of PC are bounded by the currently retreating Fourcade Glacier whose meltwater inputs, especially during summer months, significantly modify physicochemical characteristics of the inner basin, while the outer basin is less affected [37]. In this study, a transect with three sampling points, E1 (−62.232 S, −58.666 W), E2 (−62.233 S, −58.688 W), and E3 (−62.253 S, −58.714 W), was chosen. Samples were taken at two different time points, 30th of January and 14th of February 2020 from 0, 5, 10, 20, and 30 m depths, with the exception of E3, which was sampled down to 60 m depth (Table 1). Niskin bottles (10 L) were used to collect the water samples. Subsequently, 1 L of each sample was filtered through 0.22 µm Sterivex filter units (Millipore, Darmstadt, Germany) which were then filled with 70% Ethanol for sample fixation. Filters were kept at −20 °C for further DNA analysis. Physico-chemical parameters (temperature, conductivity, density and salinity) of the water column were obtained at each sampling site using a Sea-Bird SBE 19plus V2 profiler.

### 2.2. DNA Extraction and Metagenomic Sequencing

DNA was extracted according to a modified protocol described by Nercessian et al. [38]. Briefly, cell lysis was achieved using small (0.1–1 mm) zirconia-silica beads that were suspended in cetyltrimethyl ammoniumbromide (CTAB), to which anion surfactants sodium dodecyl sulfate and N-Lauroylsarcosin, proteinase K and phenol–chloroform–isoamylalcohol were added. DNA purification was facilitated by the addition of chloroform–isoamylalcohol and polyethylene glycol (PEG). Finally, DNA was precipitated at 4 °C, washed with ethanol, air-dried, and dissolved in ultra-pure water. The detailed protocol is available in Appendix A. Metagenomic sequencing was performed at Ramaciotti Centre for Genomics (Sydney, Australia). Metagenome samples were prepared for sequencing using the Illumina DNA Prep kit and sequenced on a NovaSeq 6000 platform (Illumina, San Diego, CA, USA) using v1.5 reagents. Raw metagenomic data have been deposited in the European Nucleotide Archive (ENA) (BioProject accession number PRJEB61010).

### 2.3. Sequence Processing, Metagenomic Assembly, and Binning

Raw sequence reads were adapter trimmed and filtered to remove known Illumina artifacts, PhiX, and low-quality sequences using BBDuk (v 38.18) [39]. For each sample, the clean reads were assembled de novo using SPAdes (v 3.13) [40] in metagenome mode, with option –only-assembler and longest allowed kmer length of 121, resulting in 34 single-sample assemblies. Binning of metagenome assemblies was performed for each sample by mapping reads from each sample against each of the assembled samples using BBmap [39]. To calculate the coverage of each contig within each sample, the *jgi_summarize_bam_contig_depths* was applied. Metabat2 (v 2.15) [41] was used to parse the coverage information and bin the scaffolds into genomes. Completeness and contamination of prokaryotic bins were estimated with CheckM tool (v 1.2) [42]. Following standards suggested by Bowers et al. [43], low-quality (<50% completeness, <10% contamination) and medium-quality (≥50% and <75% completeness, <10% contamination) draft metagenomes-assembled genomes (MAGs) or bins were filtered out. Taxonomic classification was performed with GTDB-Tk (v 2.1.1) [44]. High-quality MAGs were dereplicated using dRep (v 3.0) with a threshold of 98% average nucleotide identity (ANI) [45].

### 2.4. Taxonomic Profiling

Species-level molecular operational taxonomic units (mOTUs) and their relative abundances were obtained using mOTUs v3 [46].

### 2.5. Microdiversity Analysis

Microdiversity analyses were carried out using inStrain [47]. From the dereplicated set of genomes, we selected only HQ (>90% completeness, <5% contamination) representative genomes for microdiversity profiling. Quality-controlled reads from all samples were mapped to the set of dereplicated MAGs using Bowtie2 with default settings. InStrain *profile* was used to calculate the nucleotide diversity of each genome within each sample. To calculate nucleotide diversity within populations, profiles of MAGs with at least 3× coverage and 90% breadth were compared using inStrain *compare*.

### 2.6. Phylogenetic Analysis

High-quality representative MAGs of each population were used to generate phylogenetic trees using FastTree2 [48]. Trees were visualized using FigTree (v 1.4).

### 2.7. Statistical Analysis

All downstream analyses were performed in R. From 34 samples used for assembling MAGs, 9 were removed from the analysis as they represented duplicates. Taxa classified as “unassigned” were removed from the mOTU dataset. The resulting dataset was normalized using the total sum scaling method. The multivariate ordination of Principal Coordination Analysis (PCA) based on Bray–Curtis dissimilarities was used to detect patterns in community composition between sampled dates using package *phyloseq* [49] after the dataset was square root transformed. Statistical significance was tested using t.test function and adonis and pairwise.adonis functions from package *vegan*. Similarity percentage analysis (SIMPER) was used to identify MAGs that contributed most to the differences in community composition, and it was performed using package *vegan*. Data were visualized using package *ggplot*2.

## 3. Results

### 3.1. Sampling Site Characteristics

Water temperatures ranged between locations and sampling dates from 2.8 °C in the surface layers and 0.9 °C at 30 m depth (Figure 1). Generally, all stations exhibited higher temperatures on the 14th of February compared to the 30th of January. Salinity followed this trend, and all stations showed lower salinity values in the surface layer in February (Figure 1). Station E2 showed the lowest salinity at both sampling dates when compared to stations E1 and E3, while station E3 showed the highest values.

### 3.2. Bacterial Community Composition

The raw metagenomic data consisted of 3,455,125,822 sequence reads. After quality trimming and filtering, 3,030,016,014 reads remained. Metagenomic binning resulted in 3387 MAGs, including 558 high-quality (>90% completeness and <5% contamination) and 578 medium-quality (≥50% completeness and <10% contamination) MAGs. Dereplication resulted in 178 representative genomes.

Taxonomic profiling using mOTUs v3 across 34 samples identified 33,570 unique species-level mOTUs. Most abundant mOTUs were represented by HQ MAGs. Around 20–33% of reads per sample were mapped to mOTUs that were unassigned at the genus or family level and were not visualized (Figure 2A). Analysis of bacterial community composition at ranks between phyla and genus showed a relatively homogenous community with slight temporal variability (Figure 2A). Potter Cove microbial communities were dominated by mOTUs belonging to *Gammaproteobacteria*, *Alphaproteobacteria,* and *Bacteroidia* (*Bacteroidetes*). *Gammaproteobacteria* was mainly represented by the family *Oceanospirillaceae* and an unnamed family within the class. *Alphaproteobacteria* was dominated by the family *Rhodobacteraceae* and *Pelagibacteraceae,* and *Bacteroidia* (*Bacteroidetes*) was mostly represented by the family *Flavobacteriaceae*.

PCA showed a statistically significant (PERMANOVA, *p* = 0.001) separation of the samples into two clusters based on the sampling date, with the two first components explaining 35.2% of the total variation (Figure 2B). SIMPER analysis showed that the changes in the abundance of dominant taxa within the community significantly contributed to temporal variation (Table 2).

### 3.3. Intraspecific Diversity within the Community

Selecting MAGs with at least 3× coverage and 90% breadth resulted in a dataset consisting of 30 MAGs representing 21 populations. To assess the genetic variability in each population, we calculated per-sample nucleotide diversity (π), consensus ANI (conANI), and population average nucleotide identity (popANI). Nucleotide diversity is a measurement of genetic microdiversity at every position along the genome using mapped reads. popANI is a unique ANI calculation that considers both major and minor alleles. It counts single-nucleotide polymorphism (SNP) only if compared samples share no alleles. This is different from the traditional ANI (conANI in inStrain), which only considers major, consensus alleles to call a SNP. In our dataset, each population had relatively low mean nucleotide diversity (Figure 3), ranging from π = 0.00067–0.019 in January samples and π = 0.00069–0.021 in February samples. Only two populations, *Pelagibacter* and *Pseudothioglobus*, exhibited overall higher mean nucleotide diversity compared to other populations within the community (*Pelagibacter* π_January_ = 0.039, π_February_ = 0.037; *Pseudothioglobus* π_January_ = 0.028, π_February_ = 0.028). When comparing nucleotide diversity between sampled dates, two populations, UBA4582 and HTCC2207, showed significant genetic variability (*t*-test, *p* = 0.01, *p* = 0.005) with increased mean nucleotide diversity in February (Figure 3).

Of 30 MAGs, 18 showed significant differences in conANI values when compared across samples, and popANI values were always close to or above the value of 99.999%, which is considered the level at which two metagenomic populations can be reliably distinguished from one another [47]. Here, we present the results of two varying populations according to nucleotide diversity (π) values.

Gammaproteobacterial UBA4582 had a mean conANI of 99.5% within January samples and a mean conANI of 99.6% within February samples (Figure 4A). PCA analysis showed statistically significant clustering of samples by date when using conANI values (Figure 4B) (PERMANOVA, *p* = 0.001), with both axes explaining 58.94% of the total variation. popANI values were around the 99.999% threshold (Figure 4C). PCA analysis showed an overlap of the clusters, but the clustering by date was statistically significant (PERMANOVA, *p* = 0.001) (Figure 4D), suggesting differences in popANI between sampled dates.

The same pattern was observed within the Gammaproteobacterial HTCC2207.1 sp. population. Mean conANI of 99.9% was observed within both January and February samples (Figure 5A), but PCA analysis showed that conANI values of this population varied significantly across samples between the two sampling dates (Figure 5B) (PERMANOVA, *p* = 0.001) with both axis explaining 63.6% of the total variation. popANI values were above the recommended 99.999% threshold (Figure 5C). PCA analysis again showed an overlap of the clusters, whereby the clustering was statistically significant (PERMANOVA, *p* = 0.004) (Figure 4D), suggesting differences in popANI between sampled dates.

### 3.4. Phylogenetic Analysis

Sequences of closest relatives of the Gammaproteobacterial UBA4582 sp. MAG indicate a ubiquitous distribution (Figure 6) and were isolated from the North Sea (GenBank accession number, GCA_018607725.1), High-Arctic lakes (GenBank accession number, GCA_013204195.1), hydrothermal vents (GenBank accession number, GCA_012960755.1) or oxic subseafloor aquifer (GenBank accession number, GCA_016763855.1).

Gammaproteobacterial HTCC2207.1 sp. genomes clustered together mostly with sequences assembled from colder, northern regions (e.g., GenBank accession number, GCA_905182265.1, GenBank accession number, GCA_018607545.1), but also sequences isolated from hydrothermal vents (GenBank accession number, GCA_012960115.1), the Black Sea (GenBank accession number, GCA_014381985.1) or the Mediterranean (GenBank accession number, GCA_004211905.1) (Figure 7).

## 4. Discussion

In this study, we examined bacterial communities of Potter Cove and their responses to ongoing environmental changes in temperature and salinity gradients using metagenomic tools. It is known that seawater temperature plays a role in shaping bacterial communities in polar regions [12,50,51,52,53], but to what extent is still unclear. Changes in Antarctic zooplankton [54] and phytoplankton [55] communities have been observed at temperature fluctuations of less than 1 °C. Our results suggest that bacterial communities follow the same pattern.

In February 2020, weather stations measured the hottest temperature on record for the Antarctic Peninsula, reaching 18.3 °C (64.9 °F) [56,57]. The Antarctic Peninsula is a region that greatly suffers from climate change altering both structure and functioning of microbial communities at the base of Antarctic food webs. The observed extreme event of sudden temperature rise led to an increased input of glacier meltwater run-off, potentially effecting the marine ecosystem by sudden freshening. Vertical temperature profiles through the water column at our sampling locations reflected this extremely warm period (Figure 1), i.e., a pronounced increase in temperature was observed from the end of January to the mid of February. In this short period, the temperature of the entire water column, down to 30 m depth, increased by 1 °C. This temperature anomaly can also be observed when comparing the data with the previous years [58]. Furthermore, stations E1 and E2, located in the inner part of Potter Cove, exhibited warmer temperatures and a higher increase compared to outer station E3 which was the least affected part of the cove. Warming and freshening of the upper 5 m at stations E1 and E2, relative to E3, suggests that this 1 °C warming was largely a consequence of meltwater input. Warming at station E3 is likely entirely influenced by atmospheric convection and solar irradiation. We found no evidence of upwelling at any station.

Consistent with previous studies [27,59,60], *Alphaproteobacteria*, *Gammaproteobacteria,* and *Bacteroidia* (formerly *Bacteroidetes*) dominated in Potter Cove. We observed varying levels of nucleotide diversity across different populations, with *Pelgaibacter* (SAR11) and *Pseudothioglobus* as standouts. Despite these warming and freshening events being largely concentrated within the upper 5 m, the observed heterogeneity of the microbial community across the water column supports the idea that freshwater microorganisms struggle to establish in marine ecosystems [61]. Our specific analyses support this further, with mOTUs identifying taxa that were present before the warming and absent after, suggesting they were outcompeted by faster-growing taxa rather than taxa that may have been introduced in meltwaters. At the population level, we observed high popANI values indicative of a stable standing stock of diversity that was selected for or against following freshening and warming. Broadly, we argue that this freshening/warming simultaneously had a strong forcing effect at the species level, but opening new niche opportunities at the strain level for successful fast-growing microbes. As previously mentioned, we saw the loss of several taxa at the mOTUs level, and at the same time, of those populations that could be assessed between the two time-points, almost all populations exhibited slightly higher nucleotide diversity in February. Significant differences in conANI values between January and February confirm that the increase in temperature selected for specific strains within, otherwise stable population. Following warming, conANI values between samples (stations and depths) were higher, suggesting a selective pressure on a distinct genotype across the whole water column. Thus, we argue that freshening and higher temperatures in February created unfavorable conditions for several psychrotrophic taxa and stimulated rapid bacterial growth and an increase in overall nucleotide diversity for those better adapted. Despite an expansion in niche opportunities, for certain taxa, following warming at the strain level winners still emerged that tended to dominate the whole water column. Taken together, our results show that changes in community and population composition were at least partly due to in situ warming selecting from an existing genetic pool rather than advection or meltwater runoff events introducing new strains into the population. While we propose temperature as a main factor driving observed short-term succession events, we do not exclude potential roles other factors not measured in this study (i.e., viruses [62]) may play in shaping bacterial communities. In conclusion, our results indicate that inter-species genetic variants (strains) can emerge from the very rare fraction of the community due to environmental selection over short temporal scales. Populations can consist of phylotypes with differing growth optima succeeding each other and resulting in the persistence of entire populations throughout both short-term and seasonal changes [34].

Following phylogenetic analysis, we further raise questions regarding the biogeographic uniqueness of Antarctic bacteria. A ubiquitous biogeographical distribution was observed among the genomes assembled in this study. As shown in the examples of UBA4582 sp. and HTCC2207.1 sp., phylotypes were closely grouped phylogenetically with sequences isolated from other oceans, deep-sea environments, and even other ecosystems such as lakes. Vescovo et al. [27] observed a similar pattern in *Pelagibacter* sequences, proposing a potential transport of these microorganisms from the Antarctic upper water mass to temperate deep-sea basins. Freshwater ecosystems such as lakes cannot be connected to the Southern (Antarctic) Ocean through thermohaline water flows. Our phylogenetic analysis is at the whole genome level rather than based on a single marker gene. This leads us to propose that the strain similarity between organisms in distant and unconnected ecosystems may be a case of convergent evolution, where environmental drivers, in this case temperature, select for similar yet specific variants.

Consequently, understanding key factors that drive intraspecific variation of microorganisms in Antarctic waters is important for further understanding microbial responses to climate change, considering their metabolic versatility and their effect on ecosystem function.

## Figures and Tables

**Figure 1 genes-14-01051-f001:**
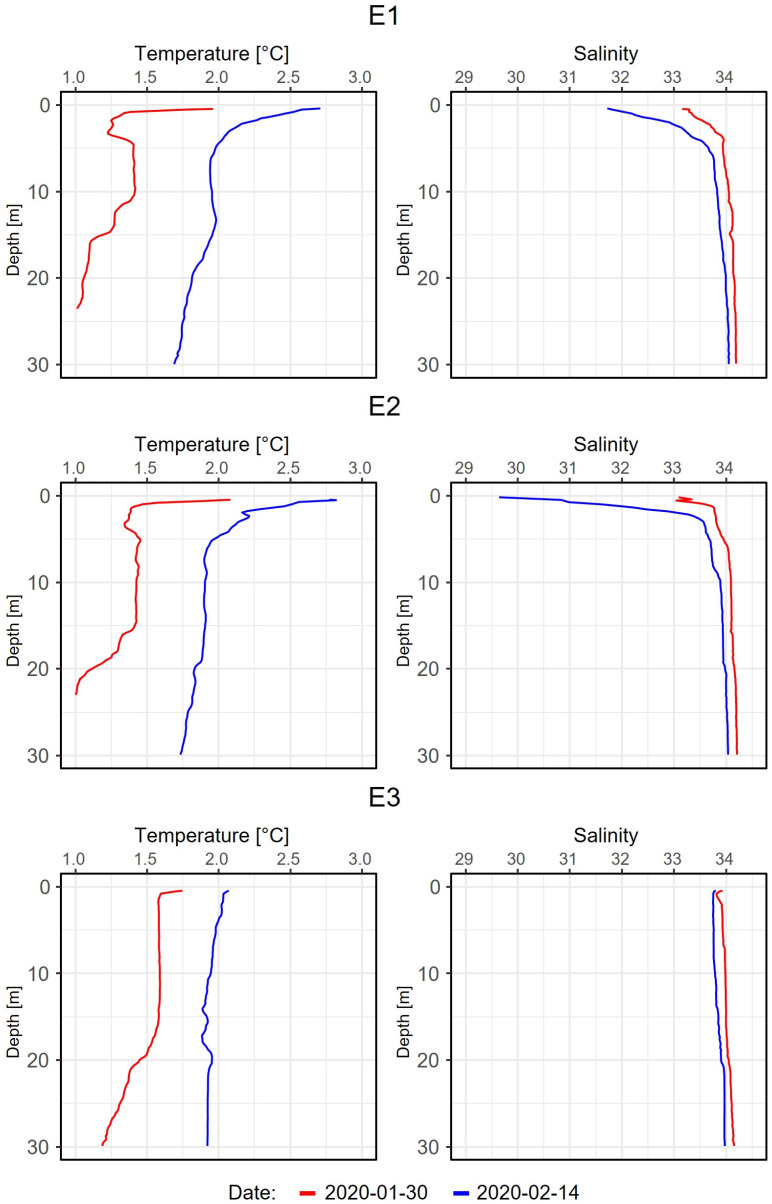
Vertical profiles of temperature and salinity obtained at stations E1, E2, and E3 on 30 January and 14 February 2020.

**Figure 2 genes-14-01051-f002:**
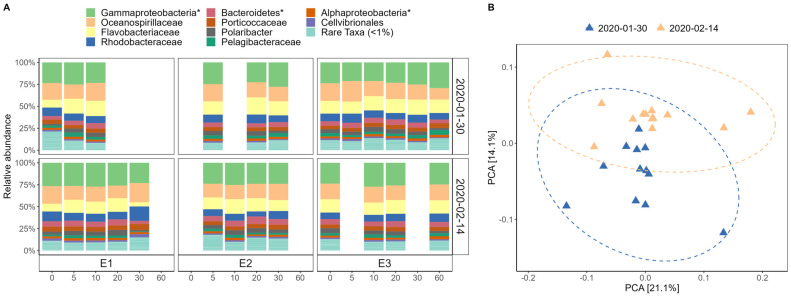
Bacterial community composition based on relative abundance of mOTUs. (**A**) community composition (unnamed family in class Gammaproteobacteria*, Alphaproteobacteria*, and Bacteroidetes*) per station, depth, and date; (**B**) PCA based on Bray-Curtis dissimilarity matrix, with confidence level for multivariate normal distribution of 95% illustrated with confidence ellipses.

**Figure 3 genes-14-01051-f003:**
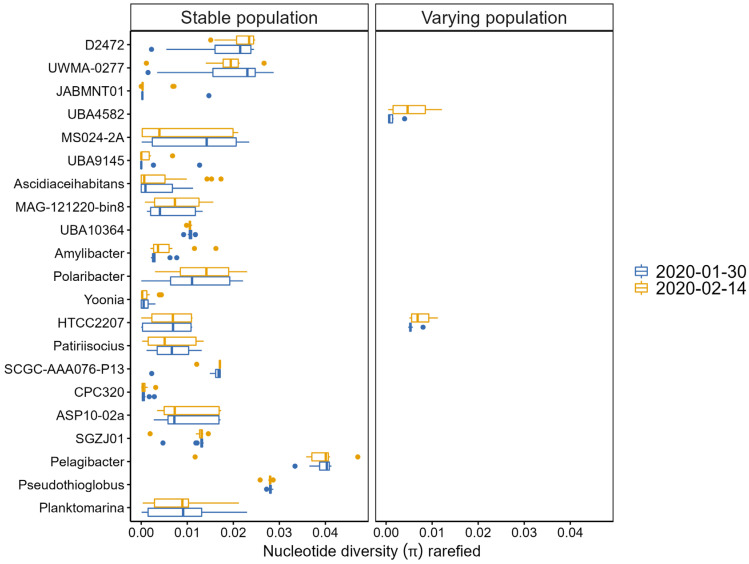
Nucleotide diversity (π) of each population compared across January and February samples. Populations that did not show statistically significant (*t*.test, *p* > 0.05) variability in nucleotide diversity between sampled dates are referred to as stable populations, and the ones that were significantly different (*t*.test, *p* < 0.05) are referred to as varying populations.

**Figure 4 genes-14-01051-f004:**
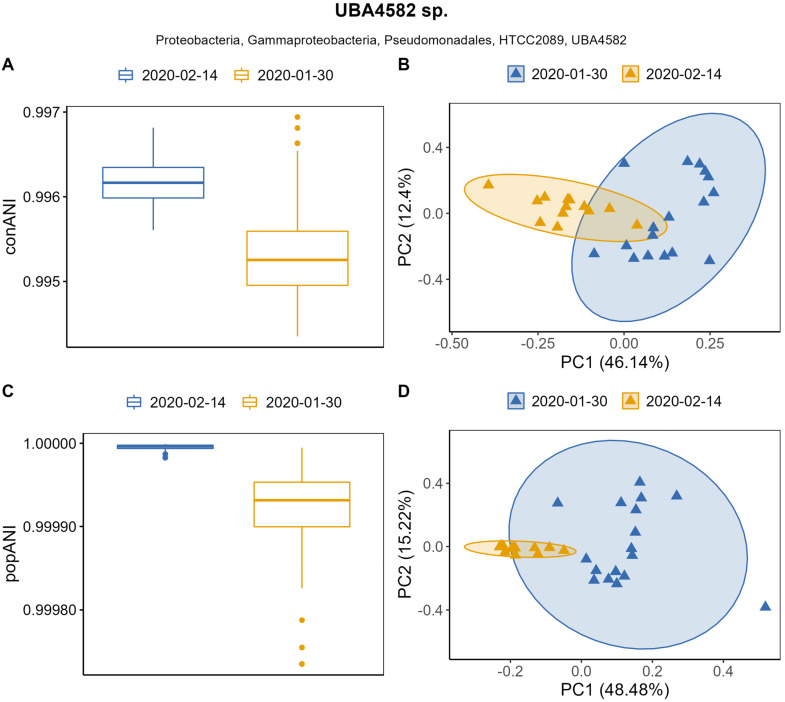
Population metrics conANI and popANI for the Gammaproteobacterial UBA4582 sp. population. (**A**) conANI between samples grouped by date; (**B**) PCA based on conANI values (PERMANOVA, *p* = 0.001) with confidence level for multivariate normal distribution of 95% illustrated with confidence ellipses; (**C**) popANI between samples grouped by sampling date; (**D**) PCA based on popANI values (PERMANOVA, *p* = 0.001) with confidence level for multivariate normal distribution of 95% illustrated with confidence ellipses.

**Figure 5 genes-14-01051-f005:**
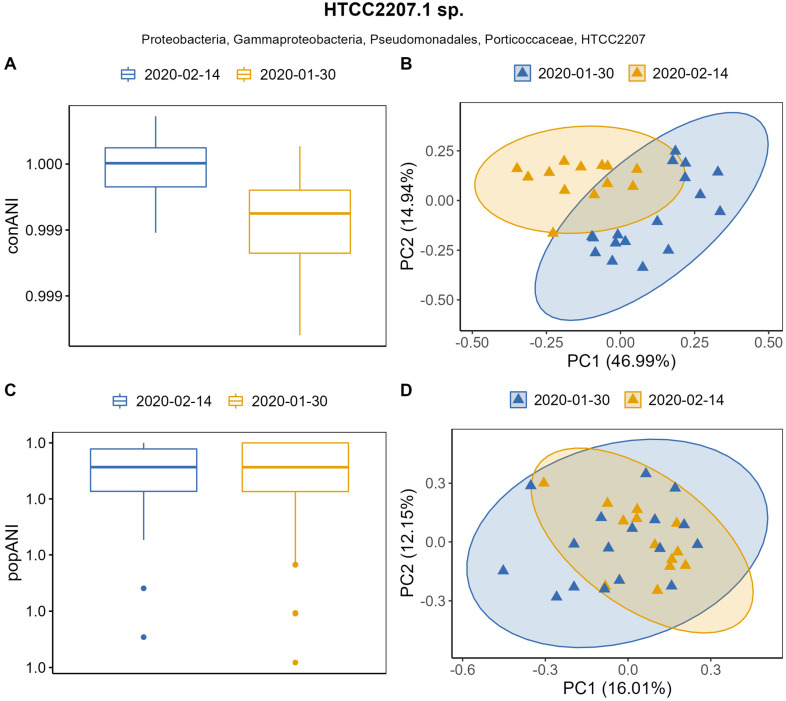
Population metrics conANI and popANI for the Gammaproteobacterial HTCC.1 sp. population. (**A**) conANI between samples grouped by date; (**B**) PCA based on conANI values (PERMANOVA, *p* = 0.001) with confidence level for multivariate normal distribution of 95% illustrated with confidence ellipses; (**C**) popANI between samples grouped by sampling date; (**D**) PCA based on popANI values (PERMANOVA, *p* = 0.004) with confidence level for multivariate normal distribution of 95% illustrated with confidence ellipses.

**Figure 6 genes-14-01051-f006:**
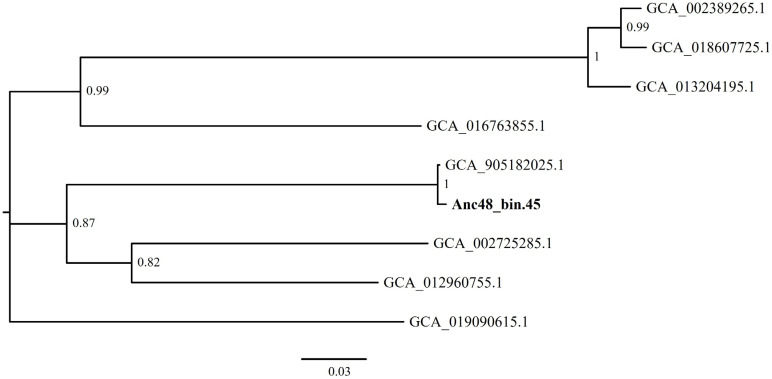
Maximum-likelihood phylogenetic tree showing positions of UBA4582 sp. phylotypes to closely related sequences in the database. The accession number of each reference sequence is given. Values at the nodes correspond to Shimodaira–Hasegawa local support test (only values of 0.5 and greater are shown).

**Figure 7 genes-14-01051-f007:**
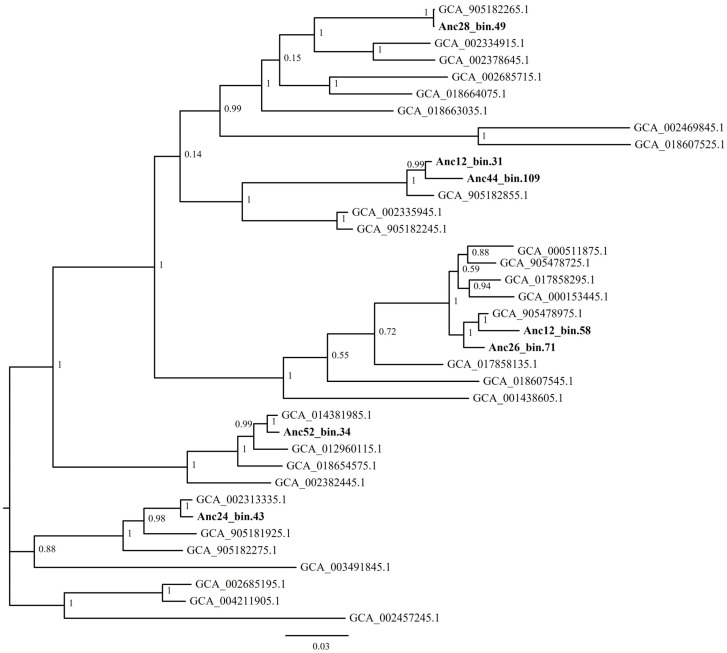
Maximum-likelihood phylogenetic tree showing positions of HTCC2207.1 sp. phylotypes to closely related sequences in the database. The accession number of each reference sequence is given. Values at the nodes correspond to Shimodaira–Hasegawa local support test (only values of 0.5 and greater are shown).

**Table 1 genes-14-01051-t001:** Samples collected and respective metadata.

Sample ID	Station	Depth (m)	Date	Temperature (°C)	Salinity (PSU)
Anc12	E1	30	14 February 2020	1.68	34.06
Anc13	E2	20	14 February 2020	1.83	34.00
Anc14	E1	20	14 February 2020	1.81	33.99
Anc17	E1	10	14 February 2020	1.95	33.83
Anc19	E2	30	14 February 2020	1.73	34.04
Anc20	E3	10	14 February 2020	1.93	33.80
Anc21	E1	0	14 February 2020	2.72	31.57
Anc22	E3	0	14 February 2020	2.06	33.78
Anc23	E3	20	14 February 2020	1.95	33.93
Anc24	E2	5	14 February 2020	1.94	33.70
Anc26	E2	10	14 February 2020	1.90	33.89
Anc28	E3	60	14 February 2020	1.32	34.17
Anc30	E1	5	14 February 2020	1.96	33.71
Anc33	E3	5	30 January 2020	1.58	33.94
Anc34	E3	30	30 January 2020	1.19	34.16
Anc35	E3	60	30 January 2020	/	/
Anc36	E1	0	30 January 2020	1.81	33.26
Anc37	E1	5	30 January 2020	1.40	33.95
Anc38	E1	10	30 January 2020	1.41	34.06
Anc42	E2	5	30 January 2020	1.45	33.98
Anc44	E2	20	30 January 2020	1.08	34.16
Anc45	E2	30	30 January 2020	0.86	34.22
Anc47	E3	20	30 January 2020	1.42	34.06
Anc49	E3	10	30 January 2020	1.59	33.99
Anc50	E3	0	30 January 2020	1.74	33.92

**Table 2 genes-14-01051-t002:** Taxa (mOTUs) contributing to similarities between January group and February group, identified by similarity percentage analyses (SIMPER). ** *p* ≤ 0.01. (unnamed family in class Gammaproteobacteria*, Alphaproteobacteria*, and Bacteroidetes*).

mOTU	Average Abundance per February Group (%)	Average Abundance per January Group (%)	Permutation *p*-Value	Contribution (%)
*Gammaproteobacteria**	4.71	3.453	0.01 **	4.6
*Oceanospirillales*	1.263	0.592	0.01 **	2.3
*Flavobacteriaceae*	1.595	2.278	0.01 **	2.6
*Rhodobacteraceae*	1.118	1.33	0.01 **	1.9
*Loktanella*	0.358	0.481	0.01 **	0.7
*Flammeovirgaceae*	0.405	0.55	0.01 **	0.7
*Cellvibrionales*	0.272	0.152	0.01 **	0.5
*Bacteroidetes**	0.601	0.521	0.01 **	1
*Bathycoccaceae*	0.377	0.202	0.01 **	0.6
*Bacteroidales*	0	0.097	0.01 **	0.3
*Flavobacteriales*	0.213	0.088	0.01 **	0.4
*Porticoccaceae*	0.106	0.08	0.01 **	0.1
*Prevotella*	0	0.032	0.01 **	0.2
*Pelagibacter*	0.018	0.043	0.01 **	0.2
*Oceanospirillaceae*	0.027	0.039	0.01 **	0.1
*Erythrobacter*	0.004	0.012	0.01 **	0.1
*Alphaproteobacteria**	0.014	0.021	0.01 **	0.1
*Lachnospiraceae*	0	0.005	0.01 **	0.1
*Brevundimonas*	0	0.003	0.01 **	0.1
*Pelagibacteraceae*	0	0.002	0.01 **	0.1
*Rhizobiales*	0	0.001	0.01 **	0.1

## Data Availability

All data are publicly available and can be found in the European Nucleotide Archive (ENA) (BioProject accession number PRJEB61010).

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
