# Peer review of "Temperature-Related Short-Term Succession Events of Bacterial Phylotypes in Potter Cove, Antarctica"

_genes, 2023, doi:10.3390/genes14051051_

Round 1

Reviewer 1 Report

Please see attached comments

Author Response

Abstract
Lines 16-17: You state, “In this study, we show that an increase in water temperature of 1 °C was enough to separate bacterial communities on a short-term temporal scale.” “Increase” suggests an in situ warming of the seawater within Potter Cove. I prefer “a change in water temperature” which is a more cautious general statement

We agree that it is a more cautious general statement, and we corrected the sentence.

Results
Lines 82-83: “In this study, we focused on the short-term temporal dynamics of bacterial communities in Potter Cove…” but only two samplings were made, 2 weeks apart. This study is based on sampling 3 stations at 3-5 depths on two separate dates two weeks apart. This resulted in 21 individual samples. This is good sample coverage for a simple comparative study, but sampling on just 2 days in one single year makes me a little concerned about the robustness of the findings. Would the same clusters appear if sampling had been done on more dates or in another year? Also, sampling at a 2 week (16 day) interval is a bit concerning since we know bacterial properties vary over 3-4 days at Palmer Station (Luria et al. 2016; Kim et al. 2016 cited in ms; and Brum et al. 2015. Seasonal time bombs: dominant temperate viruses affect Southern Ocean microbial dynamics. ISMEJ:10.1038/ismej.2015.1125.) and see also unpublished data: “Bacterial properties in discrete water column samples collected during Palmer LTER station seasons at Palmer Station Antarctica, 2002 - 2019. doi:10.6073/pasta/8168e51daba2b7f7a978d2f6a27e22a4. This makes me curious about the temporal decorrelation scale for microbial communities in these Antarctic waters. Although I would be happier about the conclusions if more sampling was undertaken, I’m well aware of the logistics difficulty of doing anything in Antarctica, let alone repeating a study over time. The authors should consider these points in the Discussion section.

In our study we do not draw conclusions regarding the year long community in Potter Cove, but rather make use of the data we obtained to draw conclusions over short term changes. Using strain-level analysis, which we improved in the revised version, we show that the change in community between the two dates is not driven by new species but rather by a change in the ratio between strains of different species. While we propose that these strains are better adapted to the changes in temperature or salinity, we cannot exclude the role of viruses in these succession events. We have now added a comment on this note to the paper.

Lines 164-65: “Of these, 3,140 had a completion level higher than 50 % and contamination level less than 10 %.” 50% genome completeness seems rather low.
I’m used to seeing values of ~70%. Ordination revealed two major clusters at the Order level with temperature being the major driver separating the clusters even though the temperature differences were slight (<=1 degree). All temperatures were above 0C and surface temperatures exceeded 2-2.5C at stations E1 and E2. Temperature, salinity and depth were the only environmental variables available for explaining the diversity and drivers among taxa. It would have been interesting to see if chlorophyll or nutrients showed up as important drivers (Kim et al). We know, for example, that bacterial activity (leucine incorporation) can be very high even at -1.8C, if chlorophyll is also high, consistent with the “Pomeroy Hypothesis” (Pomeroy, L. R. and D. Deibel. 1986. Science (Washington) 233:359-361).

Different papers report different subsets of MAGs quality. From our experience the subsets are 50-70% completeness; 70-90% completeness; and 90-100% completeness. For examples the MIMAG standards use 50-90% as medium quality and above 90% as high quality (https://www.nature.com/articles/nbt.3893).
When re-running the analysis, we increased the standards and chose MAGs with the completion level >75%, and chose only high-quality MAGs as species representatives when multiple MAGs of the same species were obtained. This is now corrected in the Materials and Methods section of the manuscript.
We fully agree with the reviewer that the analysis of more parameters would have been useful to complement the temperature, salinity, and depth measurements we have. However, these data were not collected during our sampling. Nevertheless, in our data, bacteria and eukaryotic phytoplankton (i.e. cyanobacteria and algae) did not make up any significant fraction. Therefore, we propose that at least in this case phytoplankton productivity may have not been responsible for the change in the community.

Discussion:
Lines 266-277: “In February 2020, weather stations measured the hottest temperature on record for the Antarctic Peninsula, reaching 18.3 °C (64.9 °F) [50]. The Antarctic Peninsula is a region that greatly suffers from climate change altering both structure and functioning of microbial communities at the base of Antarctic food webs. The observed extreme event of sudden temperature rise led to an increased input of
glacier meltwater run-off, potentially effecting the marine ecosystem by sudden freshening. Vertical temperature profiles through the water column at our sampling locations reflected this extremely warm period, i.e. a pronounced increase in temperature was observed from the end of January to the mid of February. In this short period, the temperature of the entire water column, down to 30 m depth, increased by 1 °C. Furthermore, stations E1 and E2, located in the inner part of Potter Cove, exhibited warmer temperatures and a higher increase compared to the more outer station which was the least affected part of the cove.”
These statements, that the increased air temperatures caused heating and increased glacial inputs within Potter Cove require some corroborating information. In order to conclude that these heating events were the result of atmospheric forcing, the authors need to show water temperature data from previous years, or cite articles demonstrating that the 2020 results are anomalous.
Hernandez et al (2014: https://link.springer.com/article/10.1007/s00300-014-1569-8) show water temperatures from station E1 and E2 used in this study. Their data shows that the temperatures measured in Feb 2020 are higher than the summer (and yearly) temperatures, while the salinity is lower than the lowest summer (and yearly) salinity for both stations.
Krock et al (2020: https://link.springer.com/article/10.1007/s00300-020-02628-z) using data from 2014-2015 provide a higher resolution temperature and salinity profiles. Over the time span discussed in our manuscript we can see changes of ca. 0.5 °C over 2 weeks. Similarly as with the Hernandez et al paper, the salinity we measured is below the scale of the data in the Krock et al., paper. According to measurements by NOAA Air temperatures for the parallel period (Jan-Feb 2014,2015 reached ad maximum of 10 °C https://www.ndbc.noaa.gov/station_history.php?station=ptcr1).

Lines 307-308: were these changes significant (p<0.05)?
Yes, PERMANOVA showed the significant effect of the temperature in this case (p=0.002). Since we now show slightly different data to better support our conclusions, we excluded this part from the manuscript. Nevertheless, all the results shown in this and the previous version are statistically significant.

Reviewer 2 Report

An analysis work was performed by Ilicic et al., which harbors some significances for the microbiological studies on the marine ecosystems. However, I some serious concerns on this work, thereby limiting the future publish process of this work.

1. I am firstly worried about the novelty of this work. Lots of works have been published for describing the microbial communities in Potter Cove. What is the difference of the current work from others? Former works obtained the very similar conclusions that salinity and temperature drive the microbial community structure in Potter Cove, and the former one conducted a more detail experiment for this conclusion.

2. Indeed, the authors stated that the isolated groups of microbes have rarely been studies for this ecosystem, as the innovation point. However, the MAGs can barely cover the entire community in the samples; therefore, I do not think the analysis result based on the MAGs is representative, which could not support the conclusion that temperature serves as a driving factor.

3. The metagenomic study information is totally missing in the method section. What is the platform? How to perform the quality control? How many data was produced? Again, I do not think the MAG community can represent the entire community, only if the authors can show the coverage of these MAGs for the entire community.

4. The statistic analysis, e.g. confidence ellipse, is missing in the PCoA figures, so how the authors conclude that the two groups could be significantly separated? Besides, the variation interpretation values are too low for a reliable dbRDA analysis.

No comments.

Author Response

1. I am firstly worried about the novelty of this work. Lots of works have been published for describing the microbial communities in Potter Cove. What is the difference of the current work from others? Former works obtained the very similar conclusions that salinity and temperature drive the microbial community structure in Potter Cove, and the former one conducted a more detail experiment for this conclusion.
Thank you for this observation. Our intention in this manuscript was not to describe the microbial community in Potter Cove and its members. Additionally, though genomic data from this area is becoming available, many of the earlier studies used metabarcoding data. Here we provide a large number of novel high-quality genomes. We harness these data to do strain resolution analysis and show that on a short temporal scale distinct changes in abiotic factors can drive intraspecific variation in this ecosystem.

2. Indeed, the authors stated that the isolated groups of microbes have rarely been studies for this ecosystem, as the innovation point. However, the MAGs can barely cover the entire community in the samples; therefore, I do not think the analysis result based on the MAGs is representative, which could not support the conclusion that temperature serves as a driving factor.
We agree with the reviewer that MAGs cannot cover the entire community. Nevertheless, neither can metabarcoding, especially when conducted at a depth of 20-50K reads per sample rather than several millions. MAGs however, offer a much higher taxonomic resolution than rRNA amplicon sequence variants (and obviously OTUs) hence they reveal other aspects of diversity not visible using single marker genes. The generation of shotgun data from which the MAGs were derived is also unbiased by primer selection hence offer a better representation of the abundances of different taxa in the water column. Last, it is the case with many analyses conducted on metabarcoding results that rare ASVs or OTUs are discarder based on different, often arbitrary criteria, narrowing the dataset to the main players (e.g. above 0.1% relative sequence frequency), thus minimizing the dataset to similar scales as that of MAGs.
The conclusion that temperature drives the short-term intraspecific succession (i.e. changes in strains within one species) is supported by statistical analyses. It is very likely that other factors that we did not measure play a role as well, since temperature does not explain the observed diversity to its full. It may also be, as reviewer 1 pointed out, and we have now added, that viruses also play a role in the community change, as was shown in the past for this environment. These viruses could as well, among others, be triggered also by temperature and thus co-vary with the change in temperature.

3. The metagenomic study information is totally missing in the method section. What is the platform? How to perform the quality control? How many data was produced? Again, I do not think the MAG community can represent the entire community, only if the authors can show the coverage of these MAGs for the entire community.
Thank you for this observation, we have modified the Materials and Methods section to add the missing information.

4. The statistic analysis, e.g. confidence ellipse, is missing in the PCoA figures, so how the authors conclude that the two groups could be significantly separated? Besides, the variation interpretation values are too low for a reliable dbRDA analysis.
Confidence ellipses have been added to PCA figures and all statistical tests with p-values performed are mentioned in the results section. Nevertheless, also in the previous version, despite the lack of graphical representation, the separation of the two communities according to date was statistically significant as was stated in the manuscript.

Round 2

Reviewer 1 Report

The authors have addressed all my points. In my opinion this ms is suitable for publication in Genes.

Reviewer 2 Report

I very appreciate the responses of the authors. However, I still adopt a cautious attitude to the representativeness of MAGs for studying the driving factor of community structure. Indeed, the metagenomics also cannot cover the whole community. But the amplicon sequencing instead of metagenomics data without amplification is often used to analyze the community structure and its driving factor. The coverage of amplicon method could reach ~99% for the whole community. Based on this coverage, the result is significant (see references 1 and 2 for example). At least, authors must show the coverage of these MAGs for the whole community.

Reference

1. Sunagawa S, Coelho L P, Chaffron S, et al. Structure and function of the global ocean microbiome[J]. Science, 2015, 348(6237): 1261359.